# ECG-Based Detection of Epileptic Seizures in Real-World Wearable Settings: Insights from the SeizeIT2 Dataset

**DOI:** 10.3390/s25247687

**Published:** 2025-12-18

**Authors:** Conrad Reintjes, Janosch Fabio Hagenbeck, Mohamed Ballo, Tim Rahlmeier, Simon Maximilian Wolf, Detlef Schoder

**Affiliations:** Cologne Institute for Information Systems, University of Cologne, 50923 Cologne, Germanyschoder@wim.uni-koeln.de (D.S.)

**Keywords:** seizure detection, machine learning, electrocardiography, anomaly detection, epilepsy, wearable sensors

## Abstract

Epilepsy is a prevalent neurological disorder where reliable seizure tracking is essential for patient care. Existing documentation often relies on self-reports, which are unreliable, creating a need for objective, wearable-based solutions. Prior work has shown that Electrocardiography (ECG)-based seizure detection is feasible but limited by small datasets. This study addresses this issue by evaluating Matrix Profile, MADRID, and TimeVQVAE-AD on SeizeIT2, the largest open wearable-ECG dataset with 11,640 recording hours and 886 annotated seizures. Using standardized preprocessing and clinically motivated windows, we benchmarked sensitivity, false-alarm rate (FAR), and a Harmonic Mean Score integrating both metrics. Across methods, TimeVQVAE-AD achieved the highest sensitivity, while MADRID produced the lowest FAR, illustrating the trade-off between detecting seizures and minimizing spurious alerts. Our findings show ECG anomaly detection on SeizeIT2 can reach clinically meaningful sensitivity while highlighting the sensitivity–false alarm trade-off. By releasing reproducible benchmarks and code, this work establishes the first open baseline and enables future research on personalization and clinical applicability.

## 1. Introduction

Epilepsy is a chronic neurological disorder characterized by the recurrent occurrence of epileptic seizures and affects approximately 0.6% of the world’s population [1]. Seizures can lead to serious medical and psychosocial consequences, e.g., falls, injuries, social isolation, anxiety, cognitive impairments, and, in extreme cases, sudden death [2]. Accurate tracking of seizure frequency is essential for individualized medication management, since it allows the assessment of effectiveness as well as the modification of dosages, and also enables an objective evaluation of the disease progression [3]. However, in clinical practice, seizure documentation is often based on patients’ self-reports, despite the well-established unreliability and incompleteness of self-reporting [4]. Hoppe et al. [5] found that less than 50% of seizures were recognized and reported by patients, while Swinnen et al. [6] observed that only about 8% of absence seizures were captured by self-reporting. Accordingly, there is a strong need for objective and automated methods for seizure detection in real-world settings.

In current research on seizure detection, two main approaches have emerged as follows: Electroencephalography (EEG)-based and Electrocardiography (ECG)-based methods. The EEG-based line of research follows the clinical gold standard, the full-scalp EEG, which directly measures neuronal activity and therefore provides the highest diagnostic accuracy [7,8]. However, this approach is hardly feasible for everyday use [9]. Consequently, recent efforts have focused on the development of wearable EEG systems, such as those explored in the SeizeIT1 [10] and SeizeIT2 [11] projects. These devices enable mobile measurements but typically cover only limited areas of the scalp, resulting in reduced detection performance [12]. Moreover, long-term use of wearable EEG-based systems (e.g., headbands or small electrodes) is often not acceptable to many people with epilepsy (PwE) due to concerns about visibility, wearing comfort, and stigma [1,13].

In parallel, a second research direction has emerged focusing on ECG-based approaches. From a practical perspective, this line of research is particularly attractive, as wearable ECG devices, ranging from medical long-term monitors to consumer-grade chest patches and smartwatches, are already widely available [14,15]. Surveys indicate that PwE are generally much more open to using such unobtrusive ECG-based wearables for seizure detection compared to EEG-based wearables [1]. Physiologically, there exists a close connection between the brain and the heart via the autonomic nervous system, allowing epileptic seizures to manifest indirectly in ECG signals [16,17]. This heart–brain interaction is clinically well established. For example, modern vagus nerve stimulation systems use heart-rate changes as a biomarker for seizure detection in FDA-approved closed-loop neurostimulation devices [18]. Thus, the ECG serves as a proxy signal: less specific than EEG, but considerably more robust and suitable for daily life, a classic trade-off between practicality and precision.

With the SeizeIT2 dataset, a publicly available dataset is now accessible for the first time that simultaneously records a two-channel EEG and a single-channel ECG in a wearable setting [11]. It represents the “first open dataset of wearable data recorded in patients with focal epilepsy” [11] (p. 1). This enables a systematic investigation of the potential of both approaches and, in particular, allows for the first evaluation of ECG-based seizure detection under real-world conditions. To our knowledge, no study has yet evaluated ECG-based seizure detection methods on this dataset. To verify this, we performed a literature search in PubMed, IEEE Xplore, and Google Scholar using the terms “SeizeIT2” and “ECG”. Because the SeizeIT2 dataset was first released in 2025, we restricted the search to publications from 2025 onward and screened titles and abstracts of the studies found.

To address this gap, we focus on the ECG signal to investigate the seizure detection potential contained in this single modality. In this work, “ECG-based seizure detection” refers to analyzing the full single-lead ECG waveform as a univariate time series. Our methods operate directly on the preprocessed ECG amplitude signal and do not extract hand-crafted heart-rate or HRV features. Instead, they detect seizure-related changes by identifying waveform segments whose temporal patterns deviate from a patient’s typical cardiac dynamics. Using this definition, we employ three different time-series anomaly-detection (AD) methods—Matrix Profile, MADRID, and TimeVQVAE-AD—to evaluate the extent to which epileptic seizures can be identified as abnormal cardiac activity. Anomaly-detection methods are particularly well suited for ECG-based epilepsy detection, as autonomic nervous system and ECG responses during seizures are highly individual and variable [19]. Classifiers often fail to capture such patient-specific patterns due to their reliance on predefined training data, whereas anomaly-detection models identify deviations from each patient’s normal heart rate behavior and are thus potentially more sensitive to previously unobserved disturbances [19]. These methods are assessed on the SeizeIT2 dataset based on sensitivity, false-alarm rate (FAR), and the Harmonic Mean Score (HMS), a composite metric that balances sensitivity and FAR through weighted penalization of false alarms.

Overall, our findings indicate that ECG-based seizure detection is feasible and that ictal changes can be captured directly from the ECG waveform. At the same time, the comparison between anomaly-detection methods highlights substantial variability in how sensitively and specifically different approaches respond to patient-specific cardiac dynamics. These general patterns underline both the potential of ECG as a practical and wearable modality and the remaining challenges related to false alarms, heterogeneous physiological responses, and the need for patient-tailored detection strategies.

Taken together, this study provides a first benchmark for ECG-based seizure detection on SeizeIT2 and motivates further research toward more robust, multimodal, and clinically scalable solutions.

The paper is organized to first outline a description of the dataset and methods, followed by a presentation of the experimental results, and a discussion of the implications, limitations, and directions for future research.

## 2. Materials and Methods

An overview of the full processing pipeline is presented in Figure 1. The workflow illustrates how raw ECG recordings from the SeizeIT2 dataset pass through quality control and standardized preprocessing before entering the model development stages. During training, each anomaly-detection method is evaluated across a broad configuration space, including model-specific parameters and post-processing strategies. In the subsequent test phase, the selected configuration is applied to unseen patient data to generate model outputs and final seizure detections.

### 2.1. Dataset

This study utilizes ECG recordings from the SeizeIT2 dataset, an open-source, multimodal wearable dataset comprising 11,640 h of physiological data from 125 patients (51 female, 41%) with focal epilepsy. Similar to the split of Bhagubai et al. [11], the validation set used to assess method performance comprises data from patients sub-001 to sub-096, while the test set includes patients sub-097 to sub-125. As noted in the paper, the patient numbering does not follow a specific order. The corpus includes 886 video-EEG-verified seizures: 317 focal aware, 393 focal impaired awareness, 55 focal-to-bilateral tonic–clonic, and 121 other/unclear events.

However, of the total 886 annotated seizures, 27 do not have corresponding ECG recordings. On the patient level, five patients (sub-048, sub-049, sub-054, sub-058, and sub-097) have no seizures with ECG recordings at all. To ensure data quality in the test set, we further examined signal saturation and excluded recordings in which at least 10% of the ECG samples were saturated. This criterion led to the exclusion of 14 recordings and reduced the total number of test seizures by three. After applying these exclusions, a total of 856 seizures with usable ECG recordings from 120 patients remained for analysis.

The data were obtained 2020–2022 across five European epilepsy monitoring centers [11]. ECG was recorded using the Sensor Dot wearable device, with electrodes placed on the left chest, extending to the lower rib cage and left parasternal region. The sampling rate was 256 Hz. All participants provided written informed consent; the original study was approved by the UZ Leuven ethics committee (ID S63631, amendment S67350).

### 2.2. Performance Metrics

Recent reviews underscore the importance of sensitivity and FAR as key benchmarks [16,20,21]. Therefore, method performance is assessed using sensitivity (recall) and FAR and the HMS. The sensitivity is calculated using any overlap between predicted and real seizure event [22] while the HMS was introduced in a SeizeIT2 seizure detection challenge and is defined as(1)HMS=Sensitivity(%) −0.4·FAR(FA/h).It combines both sensitivity and FAR into a single aggregated metric, thereby capturing the trade-off between accurate detection and alarm burden.

Statistical comparisons between models were performed using McNemar’s test for paired nominal data. For each pair of models, detection outcomes (detected vs. missed) were compared on the same seizure instances from the test set. Comparisons were conducted separately for each optimization objective (sensitivity-optimized, FAR-optimized, and HMS-optimized) and for each evaluation setting. The latter includes both the strict ictal-only detection criterion and the extended evaluation window introduced later in Section 2.4.

McNemar’s Chi-square statistic with Yates’ continuity correction was computed as(2)$$\chi^{2}=\frac{{\left(|n_{10}-n_{01}|-1\right)}^{2}}{n_{10}+n_{01}},$$
where $n_{10}$ denotes seizures detected only by model A and $n_{01}$ seizures detected only by model B. Statistical significance was assessed at α=0.05.

To quantify uncertainty in the performance estimates, we computed 95% confidence intervals by bootstrapping at the patient level. For each model, configuration, and evaluation setting (strict and SDW), we resampled patients with replacement (1000 iterations) and recomputed the patient-wise mean sensitivity, FAR, and HMS for each bootstrap sample. The 2.5th and 97.5th percentiles of the resulting empirical distributions were taken as the bounds of the 95% confidence interval. For readability and to avoid cluttering the main result tables, the confidence intervals are not included in the Results Section 3 but are reported in full in Appendix A.

Jeppesen et al. [17] showed that patients with a maximal heart rate change of more than 50 beats/min (BPM) during an seizure had significantly better performance results with ECG-based detection methods. These patients are referred to as “responders”. The distinction between responders and all patients is useful because not every patient shows pronounced autonomic changes in the ECG during a seizure [17]. This allows the actual performance of the method to be realistically evaluated and the identification of the specific patient group for whom ECG-based seizure detection is clinically appropriate.

Following Jeppesen et al. [17], we classified patients as responders if their first recorded seizure exhibited a maximum heart-rate change of more than 50 beats per minute (BPM) within a 100-RR-interval window. This responder definition was applied on a per-patient basis using the first seizure available in the SeizeIT2 recordings.

For each patient, we first detected R-peaks in the ECG using the method of Elgendi [23] and computed RR intervals for the first seizure. If this seizure contained at least 100 valid RR intervals, the maximum heart-rate change was computed directly within the seizure window. In cases where the first seizure was too short to provide 100 RR intervals, we automatically extended the analysis window symmetrically before and after the seizure, using pre-ictal and post-ictal ECG segments, until at least 100 RR intervals were available.

Using this procedure, 55.55% of patients met the responder criterion: their first recorded seizure showed a maximum heart-rate change greater than 50 BPM within the 100-RR-interval window. For this group of responders, the performance metrics are documented separately from all patients in the Results Section 3.

### 2.3. Machine Learning Methods

The time-series anomaly detection algorithms Matrix Profile, MADRID, and TimeVQVAE-AD each exhibit distinct, model-specific hyperparameters and underlying operational mechanisms. For data preprocessing, we implemented a two-fold preprocessing pipeline. First, we applied a bandpass filter to reduce the noise of the data, using a lower cutoff frequency of 0.5 Hz, a higher cutoff frequency of 40.0 Hz, and a filter order of 4. In the consecutive step, we downsampled the signal to 8 Hz to reduce the computational costs. All three anomaly-detection methods operate directly on the preprocessed ECG time series, meaning they take the amplitude of the single-lead ECG over time as input. We do not extract features such as heart rate, HRV indices, or QRS morphology. Instead, the models treat the ECG as a generic time series and flag segments whose waveform shape, envelope, or rhythm deviates from the typical cardiac pattern. While the preprocessing pipeline was used for all methods, we applied tailored post-processing steps for each model further detailed in the next subchapters.

#### 2.3.1. Matrix Profile

Figure 2 provides a concise overview of the anomaly-detection workflow based on the Matrix Profile. The method evaluates all *z*-normalized subsequences of the ECG signal and assigns each position a discord value that reflects how dissimilar the corresponding subsequence is from the similar subsequence in the time series [24]. Large peaks in the resulting Matrix Profile curve indicate discords, which are highly unusual subsequences that serve as primary anomaly candidates after global thresholding and an overlap-based filtering step. This representation forms the basis on which further refinement is applied.

To reduce the number of falsely classified seizure events produced by this pipeline, we implemented an additional post-processing step based on temporal clustering. This procedure requires a minimum of *n* consecutive detected anomalies to form a valid event, thereby mitigating false alarms caused by isolated deviations. To preserve sensitivity, short gaps between anomalies are permitted. Once a cluster is confirmed, its final anomaly index is determined as the nearest integer to the mean index of all anomalies within the group.

#### 2.3.2. MADRID

A visual overview of the MADRID workflow is provided in Figure 3. MADRID builds upon the matrix profile approach by extending anomaly detection to multiple subsequence lengths simultaneously [25]. Instead of using a fixed length *m*, discords are computed across a range of lengths $m\in [m_{\min},m_{\max}]$ [25]. The original MADRID algorithm was extended to allow for more detected anomalies, as the original implementation returned too few anomalies for successful detection. Specifically, we introduced (i) percentile-based filtering of anomalies; (ii) a flexible top-*k* selection per subsequence length; and (iii) an overlap constraint (see Figure 3). The selected values for the parameters are documented in Appendix B.

Then, MADRID returns a list of anomaly candidates. To further reduce the number of reported anomalies while preserving sensitivity, we applied a clustering step to this list. Specifically, we grouped anomalies that occurred within a short temporal window into clusters, assuming that closely occurring detections represent the same underlying event. Concretely, time-based clustering groups anomalies purely by temporal distance: for a given threshold Δt (we tested values from 2 s to 900 s), anomalies are added to the current cluster as long as the gap to the last anomaly is ≤Δt; otherwise, a new cluster is started. For each cluster, a representative anomaly is selected as the point with the minimal mean temporal distance to all others (a temporal medoid). To compare different Δt values, we defined a clustering score function that combines improvements:(3)Score=0.6·FalsePositivesReduction(%)+0.3·AnomalyReduction(%)−2.0·SensitivityLoss(%).

The procedure runs in three phases: (1) strategy evaluation: compute metrics and scores for each Δt on the training set, (2) global selection: choose the Δt with the highest average score, and (3) application: apply the selected Δt to the test set and report final metrics.

#### 2.3.3. TimeVQVAE-AD

TimeVQVAE-AD models normal temporal dynamics as a probabilistic density over discrete, time-frequency tokens and detects anomalies by low likelihood. The method has two stages followed by an evaluation/scoring [26]. A visual overview of the TimeVQVAE-AD approach is provided in Figure 4.

For stage 1, each time series $x_{\left\{1:T\right\}}$ undergoes a Short-Time Fourier Transform (STFT) preprocessing step. With fast Fourier transform size $n_{\mathrm{fft}}$, the latent height is(4)$$H=\lfloor n_{\mathrm{fft}}/2\rfloor +1.$$A convolutional encoder maps the STFT to a latent grid $z\in R^{D\times H\times W}$, where *W* is the downsampled temporal width and *D* is the number of latent channels. Vector quantization assigns each spatial location to a codebook entry from *K* prototypes, yielding a token map $s\in {\left\{1,\ldots ,K\right\}}^{H\times W}$ and its quantized embedding $z_{q}$. Stage 1 trains the encoder, vector quantizer, and decoder by minimizing the reconstruction loss.

In stage 2, a bidirectional transformer prior $p_{\theta }\left(s\right)$ is trained to predict masked tokens from surrounding context by maximizing $p_{\theta }\left(s|s_{M}\right)$, where $s_{M}$ is obtained by replacing a uniformly random subset of tokens with [MASK]. This corresponds to minimizing the negative log-likelihood of the true tokens at the masked positions. Through this process, the model learns a density over typical token configurations on the H×W grid.

For evaluation and scoring in the discrete latent space $s\in R^{H\times W}$, we slide a masking window along time. For each temporal index *w*, we mask the segment $s_{:,w-\alpha :w+\alpha }$ and compute(5)$$a_{w}=E\left[-\log p_{\theta }\;\left(s_{:,w-\alpha :w+\alpha }|s_{M}\left(:,w-\alpha :w+\alpha \right)\right)\right],$$
assigning scores only to the masked region $M^{\prime }$ via(6)$$a_{M^{\prime }}=a⊙\left(1-m\right).$$Repeating over all *w* yields $\tilde{a}\in R^{H\times W}$. To capture different durations, we repeat this for multiple α and sum the resulting $\tilde{a}$ maps.

We adapted the model with a coarse, multi-stage grid search. We first tuned stage 1 and carried the two best configurations forward into a stage 2 grid search. For selection, we used a fixed 1-h training budget and chose the model with the lowest objective: reconstruction loss for stage 1 and masked negative log-likelihood for stage 2, favoring configurations that learn efficiently on our data. We then swept anomaly-score thresholds to pick operating points optimized for specific metrics. Finally, we post-processed detections with a clustering procedure analogous to MADRID (see Section 2.3.2). Unlike MADRID, at low thresholds, TimeVQVAE-AD sometimes formed very large clusters, occasionally covering more than 50% of a recording. Therefore, we added an adaptive penalty mechanism for the evaluation of the best clustering strategies.

### 2.4. Seizure Detection Window (SDW)

Heart rate and HRV changes often precede the seizure onset by several minutes. Pavei et al. [27] quantified significant HRV changes extending up to five minutes pre-ictally. Furthermore, Jeppesen et al. [28] showed that counting an ECG-based HRV alarm as a true positive when it occurred from one minute before to three minutes after the EEG-marked seizure onset produced clinically meaningful sensitivity with an acceptable false-alarm rate.

Motivated by these findings, we define a Seizure Detection Window (SDW) ranging from –5 min to +3 min around the EEG-marked seizure onset. Any overlap of an ECG-based alarm with this SDW interval is counted as a true positive, while alarms outside this window are treated as false positives. Conceptually, the SDW captures seizure-related autonomic changes that may build up before and persist after the electrographic onset, and thus aligns the evaluation with clinically meaningful warning behavior rather than a purely ictal-only timing criterion.

To illustrate the effect of this framework in a controlled setting, Figure 5 shows synthetic ECG examples with and without SDW: without SDW, only anomalies during the ictal period are labelled as true positives, whereas with SDW, pre- and post-ictal anomalies within the [–5 min, +3 min] window are also considered true positives.

### 2.5. Seizure Type Analysis

To examine whether detection performance differs across seizure types, we conducted a seizure-type-specific sensitivity analysis based on the clinical annotations provided in the SeizeIT2 dataset [11]. The analysis was restricted to the sensitivity-optimized configurations of all three models (Matrix Profile, MADRID, and TimeVQVAE-AD).

Table 1 summarizes the seizure types represented in the test set, along with their respective sample sizes.

For each seizure type, we computed sensitivity at the level of individual seizures as the proportion of seizures correctly detected by a model within the given evaluation window (strict or SDW). We then aggregated these results in two ways: (i) as model-specific sensitivities per seizure type, and (ii) as the mean sensitivity across all three models, providing a model-agnostic estimate of relative detection difficulty across seizure types. The resulting average sensitivities are reported in the Results Section 3, whereas full per-model tables are included in Appendix C for completeness.

## 3. Results

This chapter presents the results of our conducted experiments. We begin by reporting outcomes in Section 3.1 based on a strict definition of seizure detection: a seizure is considered correctly classified only if the prediction falls within the original seizure timespan. This strict definition may not reflect practical relevance, as seizure related heart rate variability (HRV) alterations can occur beyond the original seizure timespan. An extended evaluation window is introduced in Section 2.4, and the results using it are reported in Section 3.2. For each of the three metrics: sensitivity, FAR, and HMS, we performed individual optimizations, and the best results for each metric based on the overall group are presented in the table of each subchapter. The selection was performed by filtering according to each metric on the validation set, and the corresponding values are subsequently reported on the test set.

### 3.1. Experiment Results Anomaly Detection

The results of our initial experiment, in which the strict seizure definition was applied, are presented in Table 2. We evaluate the performance using the selected metrics described in Section 2.2. Detailed information regarding the Config is provided in Appendix B.

The results show that Matrix Profile with FAR-optimized configuration achieves a sensitivity of 48.78% on the responder subset and 19.63% on the general test set, while maintaining fewer than two false alarms per hour. When adjusted to prioritize higher sensitivity, Matrix Profile achieves 90.24% sensitivity on the responder subset and 60.12% on the general test set. However, this improvement is accompanied by a substantial increase in the FAR, reaching 65.62 FA/h for the responder subset and 66.80 FA/h for the general test set.

When evaluating the MADRID model, distinct trade-offs between sensitivity and FAR become apparent. In the low–false alarm configuration, the model detects only about 2.44% of seizures in the overall dataset while generating almost no false alarms (0.11 FA/h). In contrast, tuning the model for maximum sensitivity results in a perfect detection rate of 38.1% on the responder subset and 13.4% on the overall dataset, but this comes with a notable increase in false alarms (1.47 FA/h for responders and 1.53 FA/h overall).

From the table, TimeVQVAE-AD tuned to minimize false alarms reaches 43.01% sensitivity on responders and 36.9% on the full test set, while keeping the false-alarm rate below nine per hour (8.34 FA/h and 8.58 FA/h). When tuned for higher sensitivity, sensitivity rises to 90.71% on responders and 82.79% overall, at the cost of roughly 26 FA/h (25.95 FA/h and 26.12 FA/h). As with MADRID, in this case, the HMS-optimized configuration is the same as the sensitivity-optimized one.

Statistical comparison using McNemar’s test (see Appendix Table A1) showed that, without applying the SDW, all pairwise differences between methods were statistically significant (p<0.05). In particular, the sensitivity advantages of TimeVQVAE-AD over both Matrix Profile and MADRID were highly significant in the no-SDW setting.

To complement the point estimates in Table 2, the patient-level 95% confidence intervals help contextualize the variability of the models’ performance (Appendix Table A2). For example, TimeVQVAE-AD in the sensitivity-optimized setting shows comparatively narrow CIs for sensitivity (73.42–90.28%), indicating stable performance across patients. In contrast, wider sensitivity CIs are exhibited by MADRID (11.02–31.99%) and Matrix Profile (50.71–74.34%), highlighting greater between-patient variability.

Across seizure types, substantial variability in detectability emerges when using the sensitivity-optimized configurations under the strict evaluation criterion (no SDW). As illustrated in Figure 6, focal-to-bilateral tonic-clonic seizures (f2b) show the highest average sensitivity (76.19%, n=21), indicating that these seizures exhibit the most consistent ECG abnormalities detectable by all three models. Hyperkinetic and automatisms-dominant focal impaired-awareness seizures also achieve comparatively high sensitivities (ia_m_automatisms: 68.75%, n=16; ia_m_hyperkinetic: 60.0%, n=10).

In contrast, several seizure types demonstrate only moderate detection performance: non-motor focal aware seizures (a_nm: 51.11%, n=15), non-motor focal impaired-awareness seizures (ia_nm: 47.88%, n=55), and autonomic/mixed-awareness seizures (a_um: 44.44%, n=15). The weakest performance appears for unspecific non-motor focal seizures (ua_nm: 41.67%, n=4) and hyperkinetic focal aware seizures (a_m_hyperkinetic: 36.11%, n=12). Pure focal impaired-awareness seizures without motor features (ia) achieve the lowest average sensitivity overall (26.67%, n=15).

### 3.2. Experiment Results Anomaly Detection with SDW

Table 3 presents the results of our second experiment, in which the SDW definition was applied. The same performance metrics as in the previous experiment are used for consistency. Further details on the post-processing configuration are provided in Appendix B.

As shown in Table 3, Matrix Profile with FAR-optimized configuration achieves a sensitivity of 38.04% on the full dataset and 70.73% on the responder subset, both at around 1.9 false alarms per hour. When adjusted to maximize sensitivity, performance increases to 100% on the responder subset and 98.16% on the full test set. However, consistent with the trade-off observed in Table 2, this adjustment leads to a higher FAR, reaching 13.27 FA/h for the responder subset and 13.90 FA/h for the full dataset.

When analyzing the MADRID model on the SDW extension, similar trade-offs can be observed. In the low–false alarm configuration, the model achieves 7.14% sensitivity on the responder subset and only 1.8% on the overall dataset, while producing low false alarms (0.06 FA/h for responders and 0.05 FA/h overall). In contrast, the sensitivity-focused configuration reaches 66.67% sensitivity on the responder subset and 65.24% on the overall dataset, at the cost of higher FARs (4 FA/h for responders and 4.13 FA/h overall). The HMS-optimized configuration yields the same sensitivity values of 66.67% (responders) and 65.24% (overall) while slightly reducing the FAR to 3.77 FA/h for responders and 3.96 FA/h overall.

As shown in Table 3, TimeVQVAE-AD tuned to minimize false alarms attains 59.05% sensitivity on responders and 61.09% on the full set, with 4.23 FA/h on responders and 4.22 FA/h overall. When tuned to maximize sensitivity, sensitivity rises to 100% on responders and 96.43% overall, accompanied by 40.46 FA/h on responders and 39.75 FA/h overall. Optimizing for the HMS preserves 93.33% responder sensitivity and reaches 92.86% overall, with 15.57 FA/h on responders and 15.25 FA/h overall and yields the highest HMS at 87.11 for responders and 86.76 overall.

When applying the SDW extension, most differences between models remained statistically significant (see Appendix Table A1). Notably, Matrix Profile detected a significantly larger number of seizures than MADRID across all optimization objectives (p<0.05). However, the sensitivity difference between Matrix Profile and TimeVQVAE-AD in the sensitivity-optimized configuration was not statistically significant (p=0.2482).

Confidence intervals under the SDW setting reveal a marked reduction in uncertainty for the high-performing configurations (Appendix Table A3). Matrix Profile’s sensitivity-optimized setting reaches tight CIs (88.07–100.00%), reflecting highly consistent pre-ictal and post-ictal detections across patients. Likewise, TimeVQVAE-AD shows narrow ranges in both sensitivity (89.29–100.00%) and FAR (38.34–40.82 FA/h), whereas MADRID maintains substantially wider intervals (e.g., 50.71–74.34% sensitivity), again indicating larger patient-level variability.

When applying the Seizure Detection Window (SDW), average sensitivities increase markedly across all seizure types, reflecting the extended temporal tolerance for detecting seizure-related ECG abnormalities. As shown in Figure 7, the highest detectability is observed for hyperkinetic focal impaired-awareness seizures (ia_m_hyperkinetic), which reach an average sensitivity of 96.67% (n=10). Non-motor focal aware seizures (a_nm) also achieve very high sensitivity (95.56%, n=15), followed by non-motor focal impaired-awareness seizures (ia_nm) with 90.30% (n=55), which represent the most frequent seizure type in the test set.

Focal-to-bilateral tonic-clonic seizures (f2b) reach an average sensitivity of 88.89% (n=21), while autonomic or mixed-awareness focal seizures (a_um) show similarly high detectability at 86.67% (n=15). Automatisms-dominant focal impaired-awareness seizures (ia_m_automatisms) achieve 85.42% (n=16), and hyperkinetic focal aware seizures (a_m_hyperkinetic) remain slightly lower at 83.33% (n=12). Pure focal impaired-awareness seizures (ia) reach 77.78% (n=15).

The lowest sensitivity under the SDW setting is still relatively high: unspecific non-motor focal seizures (ua_nm) achieve 75.00% (n=4), demonstrating that even seizure types with subtle manifestations benefit substantially from the extended detection window.

## 4. Discussion

### 4.1. Interpretation

The comparative evaluation of Matrix Profile, MADRID, and TimeVQVAE-AD on the SeizeIT2 dataset reveals clear differences in their performance profiles and underscores the strong influence of the evaluation framework. While the evaluated models demonstrate that ECG-based seizure detection can achieve clinically meaningful sensitivity levels, they simultaneously show that false-alarm rates remain substantially above clinically acceptable thresholds.

Under the strict seizure definition, TimeVQVAE-AD achieved the highest overall HMS (72.34) due to its favorable balance between sensitivity (82.79%) and FAR (26.12 FA/h). In contrast, Matrix Profile in its HMS-oriented configuration reached 50.92% sensitivity but at the cost of an increased FAR (21.94 FA/h), which substantially reduced its overall HMS (42.14). MADRID, while capable of operating with very low FARs, struggled with extremely limited sensitivity (13.4%) under the strict criterion, resulting in the weakest HMS of 12.8. These results indicate that the models differ not only in their raw detection capability but also in how alarms are distributed relative to annotated seizure intervals. These differences are also reflected in the statistical analysis. McNemar’s tests showed that, under the no-SDW setting, all pairwise differences between models in terms of seizure detection were highly significant (p<0.05). The bootstrapped 95% confidence intervals further support this picture (Appendix Table A2): in the FAR-optimized setting, MADRID’s FAR CI does not overlap with those of Matrix Profile or TimeVQVAE-AD. Conversely, in the HMS-optimized configuration, the HMS CI of TimeVQVAE-AD for all patients ([63.41, 79.67]) overlaps only marginally with that of Matrix Profile ([39.10, 66.64]), indicating a consistently higher overall score for TimeVQVAE-AD under the strict evaluation criterion and aligning with the significant McNemar test results.

Applying the Seizure Detection Window (−5 to +3 min relative to EEG onset) substantially altered the performance landscape and led to marked improvements across all models. Matrix Profile benefited most strongly, with sensitivity increasing from 60.12% to 98.16% and the FAR dropping to 13.9 FA/h, thereby achieving the highest HMS in the all-patient setting (92.6). TimeVQVAE-AD also improved under the SDW, reaching 92.86% sensitivity at 15.25 FA/h with a HMS of 86.76, which is highly competitive though slightly below Matrix Profile. The shift in relative ranking indicates that Matrix Profile often generates detections in the pre-ictal or post-ictal period alarms that were penalized under the strict definition but recognized as clinically meaningful within the SDW. MADRID likewise improved substantially, rising from 13.4% to 65.24% sensitivity while maintaining a relatively low FAR of 3.96 FA/h, increasing its HMS from 12.8 to 63.59. Under the SDW setting, McNemar’s tests confirmed that Matrix Profile detects significantly more seizures than MADRID across all optimization objectives (p<0.05). In contrast, the difference in sensitivity between Matrix Profile and TimeVQVAE-AD in the sensitivity-optimized configuration was not statistically significant (p=0.2482), suggesting that both models perform comparably in terms of seizure detection when SDW is applied. This interpretation is further supported by the substantial overlap of their bootstrapped 95% sensitivity CI for the sensitivity-optimized configuration (Matrix Profile: [88.07%, 100.00%], TimeVQVAE-AD: [89.29%, 100.00%]), indicating that their achievable sensitivities fall within nearly identical ranges.

When comparing operating points, distinct trade-offs become apparent. MADRID remains the model with the lowest FARs, in some cases approaching zero, but this comes at the expense of substantially reduced sensitivity. Without the SDW, TimeVQVAE-AD achieves the best overall performance, combining the highest sensitivities with comparatively moderate FARs and thus yielding the strongest HMS across all methods. When applying the SDW, Matrix Profile slightly surpasses TimeVQVAE-AD in terms of overall balance, reaching similar sensitivities but slightly higher HMS values, although the difference in sensitivity between these two models is not statistically significant according to McNemar’s test (p=0.2482). Together, these operating regimes delineate a clear trade-off curve: MADRID represents the low-alarm regime, TimeVQVAE-AD dominates under strict evaluation criteria (no SDW), and Matrix Profile leads in the extended detection framework (with SDW), highlighting that different models may be preferable depending on the chosen evaluation window and clinical tolerance for false alarms.

The responder analysis further highlights substantial inter-individual variability. Under the SDW, both Matrix Profile and TimeVQVAE-AD achieved 100% sensitivity in responders, but Matrix Profile at more moderate alarm rates around 13.27 FA/h, while TimeVQVAE-AD had 40.46 FA/h. In contrast, the all-patient sensitivity was consistently lower, reflecting the presence of non-responders whose ECG signatures showed weaker or atypical ictal dynamics. Our findings are consistent with prior research showing that ECG-based seizure detection methods perform better on responders (patients with a maximum heart rate change of >50 BPM during a seizure) [17,29]. Furthermore, the divergence between responder and all-patient performance suggests that personalization strategies, such as patient-specific thresholds or adaptive retraining, will be essential to lift performance in non-responders without over-alarming responders.

This interpretation is further supported by the CI reported in the Appendix A. Several of the sensitivity and HMS intervals remain relatively broad, reflecting substantial dispersion in patient-wise performance. Because these intervals are derived from patient-level sensitivity estimates, their width highlights the pronounced heterogeneity in how seizures are detected across individuals in the SeizeIT2 cohort. This variability underscores the importance of personalization strategies.

Beyond inter-individual variability, a second source of heterogeneity arises from differences between seizure types themselves. Under the strict evaluation criterion, seizure types characterized by pronounced motor activity or marked autonomic involvement such as focal-to-bilateral tonic-clonic seizures or hyperkinetic impaired-awareness seizures exhibited the highest sensitivity, whereas non-motor and subtle impaired-awareness seizures remained substantially more difficult to identify. This pattern aligns with the expectation that stronger sympathetic activation and movement-related artifacts produce clearer ECG anomalies, but it partially matches prior ECG-based findings. Vandecasteele et al. [30] reported that focal aware (a) and focal-to-bilateral tonic-clonic (f2b) seizures were detected more reliably on only ECG data than focal impaired-awareness (ia) seizures, whereas in our analysis, impaired-awareness seizures with prominent motor features are among the best-detected types, while non-motor ia events remain challenging. At the same time, both studies consistently identify focal-to-bilateral tonic-clonic seizures as one of the most reliably captured types in ECG-based detection. This broader trend is further supported by HRV-based work: Nouboue et al. [31] reported higher sensitivity for convulsive than for non-convulsive seizures, and Chen et al. [32] found more frequent and pronounced ictal tachycardia in complex-partial (=ia) and secondarily generalized seizures (=f2b) compared to simple-partial events (=a).

When applying the SDW, however, sensitivities increased across all seizure types, with several categories exceeding 90%. This uniform improvement suggests that many seizure-related ECG changes occur outside the narrowly defined EEG onset window, particularly in seizure types with gradual autonomic buildup or post-ictal cardiac effects. Taken together, these findings indicate that ECG-based detection is most effective for seizures with robust autonomic signatures, while the SDW framework mitigates some of the challenges inherent to more subtle seizure types by capturing clinically relevant pre- and post-ictal cardiac dynamics.

A broader physiological perspective helps contextualize these findings, as the ECG anomalies detected by our models ultimately arise from generic autonomic and cardiophysiological responses. Previous studies have demonstrated that epileptic seizures are often accompanied by characteristic changes in cardiac activity, including alterations in ECG waveform morphology [33,34]. These effects are generally attributed to rapid shifts in autonomic regulation, where seizures can briefly increase sympathetic drive or reduce vagal influence [35]. Such autonomic responses can lead to measurable changes in heart rate as well as in specific ECG segments [35,36]. Our findings are consistent with this line of research, as the evaluated models capture several of these atypical seizure-related cardiac patterns as anomalies. At the same time, the approaches used here do not aim to explain the underlying physiology in full detail, but rather to make use of the observable ECG dynamics that reliably co-occur with many seizures.

Bhagubai et al. [11] also evaluated two models on the SeizeIT2 dataset using EEG data, a Support Vector Machine (SVM) and the deep learning architecture ChronoNet. Similar to our results, their findings illustrate the same fundamental trade-off between sensitivity and false-alarm rate: while ChronoNet achieved the highest sensitivity reported in their study (84.2%), this operating point corresponded to 100.5 FA/h. The SVM reached a lower sensitivity (71.1%) with substantially fewer false alarms (11.0 FA/h).

In contrast, our experiments using only ECG data show that the TimeVQVAE-AD model, as our best performing model without the SDW extension, reaches a comparable sensitivity of 82.79% while producing nearly four times fewer false alarms per hour than ChronoNet (26.12 FA/h vs. 100.5 FA/h). This relationship is illustrated in Figure 8, which directly compares sensitivities and false-alarm rates of EEG-based and ECG-based approaches. A direct comparison with the SVM baseline is less straightforward, as none of the ECG or EEG models operate at a similar sensitivity-to-false-alarm ratio. When optimizing TimeVQVAE-AD for minimal false alarms (approximately 8 FA/h), sensitivity decreases to 36.9%, indicating that future work could explore an SVM-based approach applied to ECG data.

While some earlier studies working exclusively with ECG data have reported lower FAR values [16,29,37], these results were often achieved on smaller or more curated datasets. Seth et al. [21] found that in their systematic review, none of the reviewed ECG-only studies have been validated on cohorts exceeding 43 patients. In contrast, the SeizeIT2 dataset encompasses a broader and more heterogeneous patient population, which introduces additional variability but also better reflects real-world conditions. In line with calls in the literature for more open benchmark datasets [38,39], we encourage the community to further explore SeizeIT2, develop new approaches, and share results to support transparent comparison and collective progress in seizure detection research.

Building on this perspective, the use of large, open datasets such as SeizeIT2 can be seen as a key enabler for scalable clinical translation. ECG sensors are already integrated into many wearables and hospital monitors, making ECG-based approaches attractive for low-burden, widely deployable seizure monitoring. Although the achieved FARs remain too high for autonomous clinical deployment (goal of 90% sensitivity and 2 FA/week (≈0.01 FA/h)) [40], the sensitivity levels reached in this work already enable clinician-assisted workflows. For example, an ECG-based anomaly detector embedded in a wearable and linked to a smartphone application could screen the ECG in the background, notify the user or caregiver about suspicious events, and store them as an objective seizure diary that can be reviewed during consultations. In this scenario, clinicians would receive preselected segments and summary statistics rather than raw continuous ECG, which could support therapy adjustments, remote follow up, and shared decision making. From the user perspective, such a wearable would provide a discreet and always available companion that helps to document seizures more reliably. Moreover, the strong responder performance indicates that ECG-based monitoring could be particularly useful for patients with pronounced autonomic changes during seizures, where personalized models could be calibrated to the individual wearable signal over time. Together, these elements outline a realistic pathway for integrating ECG-based seizure detection into wearable devices and clinical workflows once false alarms can be further reduced. In this sense, the presented models should not be interpreted as standalone diagnostic tools, but as a first benchmark that demonstrates the potential of ECG only anomaly detection on SeizeIT2 and motivates future work on wearable centred applications.

### 4.2. Limitations and Further Research

This study has several limitations, which at the same time highlight promising directions for future research.

Our optimization for TimeVQVAE-AD was computationally constrained as follows: we used a coarse grid search and capped training at a fixed 48 h budget per stage. As a result, some models likely did not fully converge, and longer training or a broader search may yield stronger performance. In the evaluation, we also set the rolling window stride to the smallest value we could afford computationally, rather than the theoretical minimum of a one-sample shift. While this ensured computational tractability, an even smaller stride could provide denser coverage and potentially better accuracy.

Furthermore, we did not systematically evaluate the real-time capabilities of the methods. The analysis of the dataset required several days of computation across all methods, indicating that substantial optimization is necessary before practical deployment becomes feasible. Future research should therefore aim to improve computational efficiency and systematically investigate whether these anomaly-detection approaches can achieve real-time performance.

Our seizure type analysis shows that some seizure types are detected considerably more reliably than others across models. Although these findings highlight meaningful differences in detectability, some seizure types are represented by only a small number of events, with the smallest group comprising just four seizures. This constrains the robustness of our overall assessment of detection reliability, especially for seizure types with low sample sizes. Furthermore, we did not investigate the underlying physiological or algorithmic causes in detail. Future work could explore these mechanisms more systematically and assess whether model adaptations or even type-specific detection strategies such as training dedicated models for selected seizure types may further improve performance.

In addition, the proposed methods were based solely on ECG data rather than feature-based or multimodal anomaly detection, even though the dataset also provides additional physiological modalities like EEG. While effective, this approach excludes the potential benefits of domain-specific features such as HRV or other ECG-derived biomarkers, as well as the complementary information that could be gained from combining multiple biosignals. Future research could therefore explore hybrid or multimodal approaches.including the incorporation of HRV features or arrhythmia indicators, which may capture seizure-related cardiac phenomena beyond the coarse waveform dynamics analyzed in the present study.

A direct comparison with earlier ECG detection approaches was not feasible within our evaluation setup. In general, such comparisons are limited either by fundamental methodological differences, most notably the use of personalized training schemes instead of evaluation on unseen patients as required in the SeizeIT2 framework, or by the lack of publicly available implementations, which hampers reproducibility. Nevertheless, prior work has shown that personalization strategies can substantially improve sensitivity and reduce false alarms [29,41]. These findings make personalization an interesting avenue for future research; however, it is important to recognize that such approaches operate under fundamentally different conditions than the models evaluated here, which were assessed strictly on previously unseen patients. Incorporating personalized methods into the SeizeIT2 benchmark would therefore require a dedicated extension of the evaluation design and should be explored in future studies.

## 5. Conclusions

This study provides the first systematic benchmark of ECG-based seizure detection on the large-scale SeizeIT2 dataset using three advanced anomaly-detection methods. By analyzing more than 11,000 h of wearable ECG recordings with 886 video-EEG-verified seizures, the evaluation yields a more realistic estimate of expected performance in everyday real-world application.

The results demonstrate that seizure-wise sensitivities exceeding 90% are achievable. Under the SDW, Matrix Profile achieved 98.16% sensitivity at 13.9 false alarms per hour with a HMS of 92.60, while TimeVQVAE-AD reached 92.86% sensitivity at 15.25 false alarms per hour with a HMS of 86.76. MADRID showed a different profile, with lower sensitivity (65.24%) but substantially fewer alarms (3.96 FA/h), indicating that it can be tuned for alarm reduction at the cost of missed detections. These findings highlight the existence of distinct operating regimes: Matrix Profile provided the most favorable balance between sensitivity and false alarms under the SDW, TimeVQVAE-AD dominates without the SDW, and MADRID offered low-alarm configurations with moderate sensitivity.

Despite these encouraging sensitivities, the observed FAR remain far above levels acceptable for practical use. In real-world scenarios, user acceptance typically requires fewer than two false alarms per week, whereas the best-performing models in this study still generated more than 10 FA/h. This discrepancy indicates that current implementations are not yet suitable for stand-alone deployment. However, they can be used as pre-screening tools.

In conclusion, this work establishes a reproducible baseline for ECG-based seizure detection on a large open dataset and underscores both the promise and the limitations of anomaly-detection methods. Future research should prioritize strategies for false alarm reduction, patient-specific adaptation, and multimodal signal integration in order to bridge the gap between algorithmic performance and the requirements of real-world applicability.

## Figures and Tables

**Figure 1 sensors-25-07687-f001:**
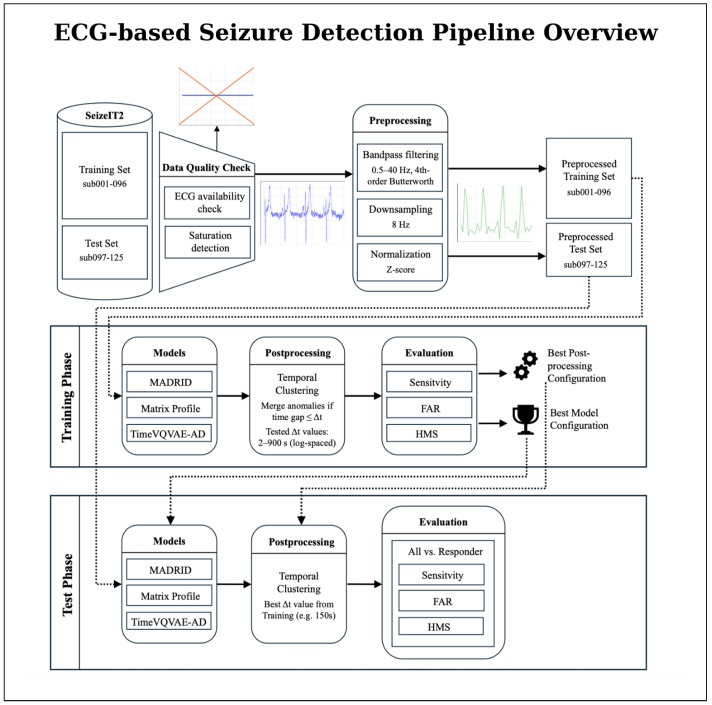
Overview of the proposed ECG-based seizure detection pipeline using the SeizeIT2 dataset. The overall workflow is divided into a training phase and a test phase. The dataset is first split at the patient level into a training set (sub-001–096) and a test set (sub-097–125). ECG recordings undergo data quality checks (ECG availability, and saturation detection) and preprocessing (bandpass filtering, normalization, and downsampling), resulting in preprocessed training and test sets. In the training phase, all three anomaly-detection models (Matrix Profile, MADRID, and TimeVQVAE-AD) are trained and evaluated across multiple configurations, followed by post-processing via temporal clustering. Based on training performance, the best model and post-processing configuration are selected. In the test phase, this selected configuration is applied to the held-out test patients, and seizure events are detected and evaluated. Performance metrics are reported separately for all patients and for the subgroup of responders exhibiting pronounced ictal heart-rate changes.

**Figure 2 sensors-25-07687-f002:**
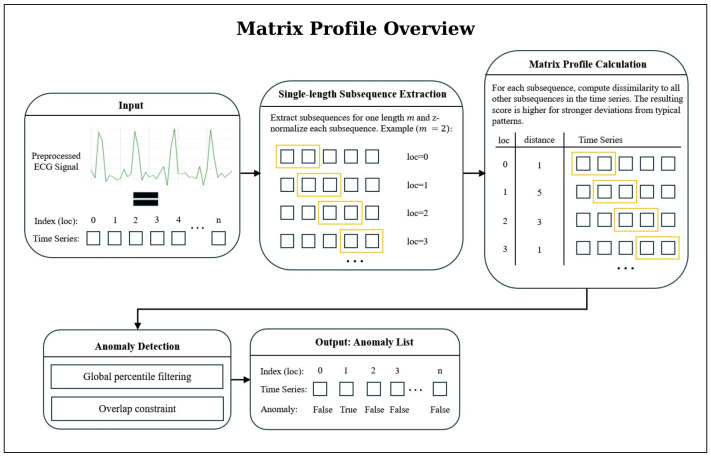
Overview of the Matrix Profile-based anomaly-detection workflow. From a preprocessed ECG signal, all *z*-normalized subsequences of length *m* are extracted and compared using the Matrix Profile, which assigns to each position the distance to its nearest neighbor subsequence. Large values mark discords (candidate anomalies) which are then filtered through percentile thresholding and an overlap criterion to remove redundant or weak candidates.

**Figure 3 sensors-25-07687-f003:**
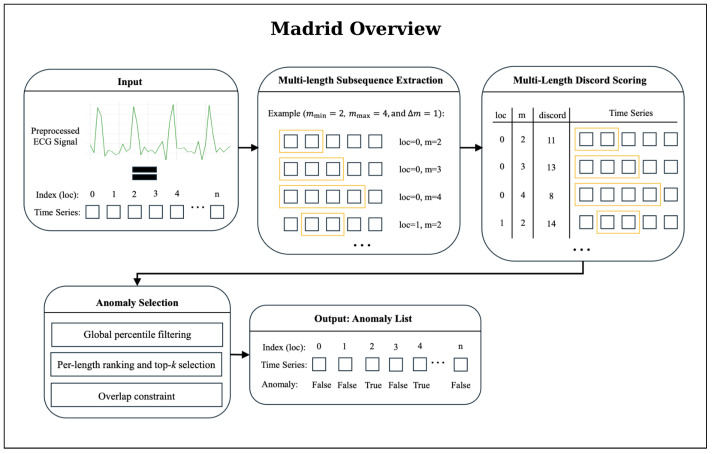
Overview of the MADRID anomaly-detection workflow. The algorithm first extracts subsequences for multiple lengths *m*, ranging from $m_{\min}$ to $m_{\max}$ in steps of Δm (step size). For each subsequence, MADRID computes a discord score that quantifies the degree of deviation from typical patterns in the time series; higher scores indicate stronger dissimilarity. The resulting multi-length discord matrix is refined through a three-stage anomaly-selection procedure: (1) Global percentile filtering: subsequences below a global percentile threshold are removed, ensuring that only strongly deviant patterns remain. (2) Per-length ranking and top-*k* selection: subsequences are ranked by discord score for each length and the top *k* candidates are retained. (3) Overlap constraint: candidates overlapping more than 25% with already selected anomalies are discarded, yielding a final set of non-redundant anomaly candidates. This filtered anomaly list forms the basis for post-processing, clustering, and seizure-detection evaluation.

**Figure 4 sensors-25-07687-f004:**
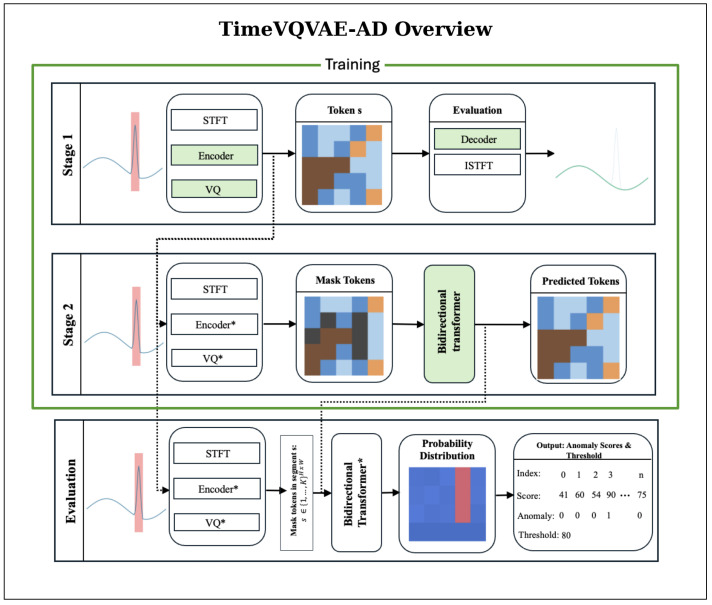
TimeVQVAE-AD overview. During training (Stages 1 and 2), only normal intervals are used; anomalous regions are excluded. Stage 1 (Tokenizer): Normal segments are transformed by the STFT and passed through the encoder and vector quantizer to obtain the token map $s\in {\left\{1,\ldots ,K\right\}}^{H\times W}$. The encoder, vector quantizer, and decoder are optimized with a reconstruction objective; iSTFT of the decoded STFT yields an approximate waveform. Green boxes denote modules optimized in this stage. Stage 2 (Prior): The tokenizer is frozen. From the same normal-only data, randomly chosen windows *M* are masked and a bidirectional transformer prior is trained to predict masked tokens by maximizing $p_{\theta }\left(s|s_{M}\right)$, equivalent to minimizing masked negative log-likelihood. Green boxes denote modules optimized in this stage. Evaluation: With the tokenizer and prior fixed, a masking window of width α slides along time. For each center index *w*, the prior outputs categorical distributions that quantify how likely the masked tokens are under the learned normal model; the anomaly score is the mean negative log-likelihood over the masked span. Repeating over *w* yields a score map $\tilde{a}$. Scores across multiple α may be summed, and a threshold τ determines anomaly labels.

**Figure 5 sensors-25-07687-f005:**
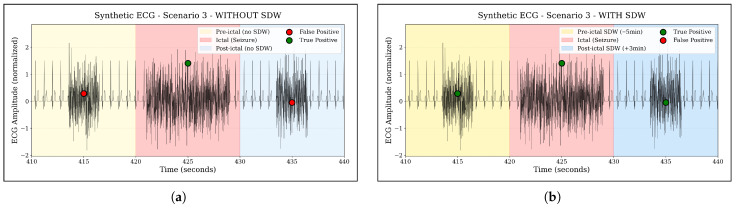
Illustration of the Seizure Detection Window (SDW) concept on a synthetic ECG example. The colored regions mark the ictal period and the extended SDW interval from five minutes before to three minutes after seizure onset. Anomalies occurring outside the SDW are treated as false positives, whereas anomalies within the SDW are counted as true positives, reflecting clinically meaningful pre- and post-ictal autonomic changes. (**a**) Without SDW: only ictal anomalies are counted as true positives. (**b**) With SDW: pre- and post-ictal anomalies inside the –5 min, +3 min window are also counted as true positives.

**Figure 6 sensors-25-07687-f006:**
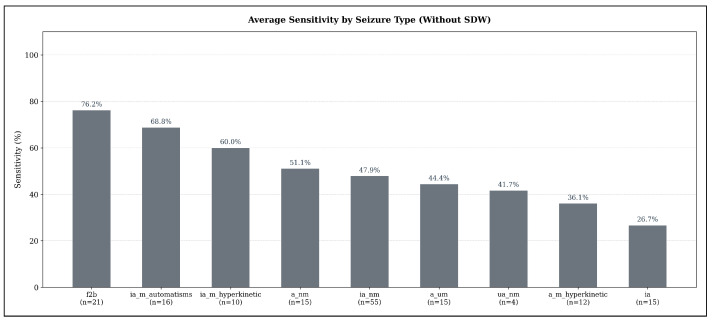
Average seizure-type-specific sensitivity across all models in the sensitivity-optimized configuration, evaluated under the strict detection criterion (no SDW). Sensitivity values represent the mean sensitivity across TimeVQVAE-AD, Matrix Profile, and MADRID; numbers below each bar indicate the number of seizures of that type in the test set.

**Figure 7 sensors-25-07687-f007:**
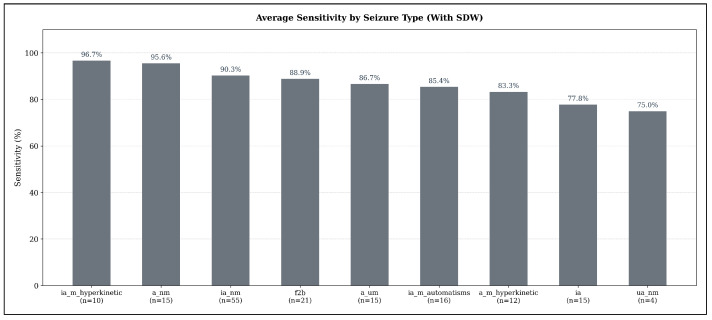
Average seizure-type–specific sensitivity across all models in the sensitivity-optimized configuration, evaluated under the expanded detection criterion (SDW). Sensitivity values represent the mean sensitivity across TimeVQVAE-AD, Matrix Profile, and MADRID; numbers below each bar indicate the number of seizures of that type in the test set.

**Figure 8 sensors-25-07687-f008:**
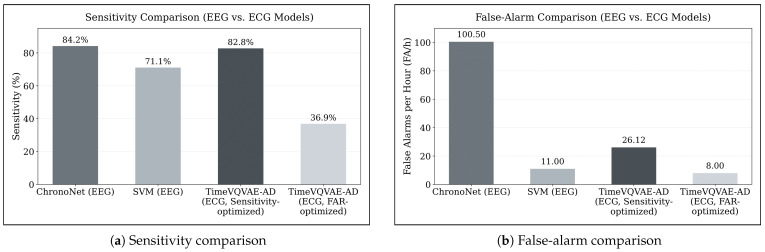
Comparison of EEG-based and ECG-based seizure detection performance on the SeizeIT2 dataset. (**a**) Sensitivity of ChronoNet and SVM (EEG-based baselines from Bhagubai et al. [11]) compared with TimeVQVAE-AD evaluated on ECG data in its sensitivity-optimized and false-alarm-optimized configurations. (**b**) Corresponding false-alarm rates (FA/h) for the same models. The figure highlights the fundamental trade-off between sensitivity and false-alarm rate and demonstrates that ECG-based TimeVQVAE-AD can achieve sensitivity comparable to ChronoNet while producing substantially fewer false alarms.

**Table 1 sensors-25-07687-t001:** Seizure types included in the test set and their frequencies.

Seizure Type (Expanded Description)	n
Focal aware, unimpaired awareness, non-motor (FA-UA-NM)	4
Focal impaired awareness (FIA)	15
Focal to bilateral tonic-clonic (F2B)	21
Focal aware, unspecified motor signs (FA-A-UM)	15
Focal aware, non-motor (FA-A-NM)	15
Focal aware, motor hyperkinetic (FA-A-M-Hyperkinetic)	12
Focal impaired awareness, motor automatisms (FIA-M-Automatisms)	16
Focal impaired awareness, motor hyperkinetic (FIA-M-Hyperkinetic)	10
Focal impaired awareness, non-motor (FIA-NM)	55

**Table 2 sensors-25-07687-t002:** Performance metrics of Matrix Profile, MADRID, and TimeVQVAE-AD optimized for different objectives without SDW applied. Sensitivity, FAR, and HMS are reported for both responders and all patients. Boldfaced values indicate the best performance within each metric and test set. The *p*-value indicates the statistical difference in the All-metric compared to the best-performing model for the respective optimization objective (as marked in bold).

Method	Optimized for	Sensitivity (%)		FAR (FA/h)		HMS		*p*-Value
		Responder	All	Responder	All	Responder	All	
Matrix Profile	FAR	48.78%	19.63%	1.92	1.92	48.01	18.86	<0.0001
MADRID	FAR	9.52%	2.44%	**0.11**	**0.11**	9.48	2.40	–
TimeVQVAE-AD	FAR	43.01%	36.90%	8.34	8.58	39.76	33.46	<0.0001
Matrix Profile	Sensitivity	90.24%	60.12%	65.62	66.80	63.99	33.40	<0.0001
MADRID	Sensitivity	38.10%	13.40%	1.47	1.53	37.51	12.80	<0.0001
TimeVQVAE-AD	Sensitivity	**90.71%**	**82.79%**	25.95	26.12	80.33	72.34	–
Matrix Profile	HMS	87.80%	50.92%	21.22	21.94	79.32	42.14	<0.0001
MADRID	HMS	38.10%	13.40%	1.47	1.53	37.51	12.80	<0.0001
TimeVQVAE-AD	HMS	90.71%	82.79%	25.95	26.12	**80.33**	**72.34**	–

**Table 3 sensors-25-07687-t003:** Performance metrics of Matrix Profile, MADRID, and TimeVQVAE-AD optimized for different objectives with SDW applied. Sensitivity, FAR, and HMS are reported for both responders and all patients. Boldfaced values indicate the best performance within each metric and test set. The *p*-value indicates the statistical difference in the All-metric compared to the best-performing model for the respective optimization objective (as marked in bold).

Method	Optimized for	Sensitivity (%)		FAR (FA/h)		HMS		*p*-Value
		Responder	All	Responder	All	Responder	All	
Matrix Profile	FAR	70.73%	38.04%	1.91	1.90	69.97	37.28	<0.0001
MADRID	FAR	7.14%	1.80%	**0.06**	**0.05**	7.12	1.78	–
TimeVQVAE-AD	FAR	59.05%	61.09%	4.23	4.22	57.36	59.40	<0.0001
Matrix Profile	Sensitivity	**100.00%**	**98.16%**	13.27	13.90	94.69 ^1^	92.60 ^1^	–
MADRID	Sensitivity	66.67%	65.24%	4.00	4.13	65.07	63.59	<0.0001
TimeVQVAE-AD	Sensitivity	**100.00%**	96.43%	40.46	39.75	83.82	80.53	0.2482
Matrix Profile	HMS	97.56%	96.93%	10.92	11.37	**93.19**	**92.39**	–
MADRID	HMS	66.67%	65.24%	3.77	3.96	65.16	63.65	0.0056
TimeVQVAE-AD	HMS	93.33%	92.86%	15.57	15.25	87.11	86.76	<0.0001

^1^ For Matrix Profile, the sensitivity- and HMS-optimized configurations were nearly identical on the training set, which can cause the sensitivity-optimized setting to yield slightly higher HMS on the test data.

## Data Availability

The data presented in this study are openly available in OpenNeuro at https://doi.org/10.18112/openneuro.ds005873.v1.1.0 (accessed on 17 March 2025), reference number ds005873. The source code is available via a GitHub repository: https://github.com/creintjes/ecg-seizure-detection.

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
