# Peer review of "ECG-Based Detection of Epileptic Seizures in Real-World Wearable Settings: Insights from the SeizeIT2 Dataset"

_sensors, 2025, doi:10.3390/s25247687_

Round 1
Reviewer 1 Report
Comments and Suggestions for Authors
Recommendation: MAJOR REVISION
Justification: This is a well-executed study that makes significant contributions to the field of wearable seizure detection. The experimental design is rigorous, the evaluation is comprehensive, and the results are presented transparently. The open release of code and reproducible benchmarks will enable substantial future research. However, several areas require clarification or expansion before publication.
------------------------------------------------
This manuscript represents a significant contribution to the field by establishing the first comprehensive ECG-based seizure detection benchmark on the SeizeIT2 dataset. Although the experimental design is robust and the reporting transparent, the current presentation contains critical gaps that necessitate substantial revisions to permit a full scientific assessment. To address these issues:
- It is imperative to augment the manuscript with figures detailing the system pipeline, detection instances, sensitivity-FAR tradeoffs, and the SDW framework.
- The statistical rigor in Section 2.2.2 must be improved through the inclusion of significance testing and the rectification of confidence intervals and p-values within the primary tables.
- Authors should implement a comparative analysis against HRV-based detection baselines, such as Jeppesen’s method, and provide comprehensive computational specifications regarding hardware and real-time assessment capabilities in Section 2.4.
- Structural and analytical improvements are also required, including a deeper analysis of seizure types, the reduction of the abstract focusing on significant results, the relocation of the SDW definition to the Methods section, clarification of the responder window extension procedure, and a more thorough discussion of the potential for clinical translation.
Reviewer 2 Report
Comments and Suggestions for Authors
I would like to thank the authors for their very interesting study and the important topic selected as the main aim of this study. This is likely one of the few studies aimed to detect the epileptic seizures based on ECG abnormalities detection based on fairly large public databases. The research task is very relevant, especially in modern conditions. It is entirely consistent with the modern trend of expanding the scope of ECG application, primarily through the use of wearable miniature ECG devices.
The article is written quite clearly and consistently, the statistical treatment is generally modern and convincing.
Nevertheless, there are some concerns after reading the manuscript that are listed below.
The article is titled "ECG-Based Detection of Epileptic Seizures." Naturally, one would expect a fairly detailed description of the ECG phenomena that are signs of an epileptic episode. However, while reading the article, I still didn't understand exactly what ECG changes are being discussed. The general term "ECG anomaly detection" is used, but it's certainly not descriptive enough. I assume they're talking about certain HRV parameters (which ones exactly?) and simply heart rate. Then perhaps the article should be titled "HRV and HR Detection of Epileptic Seizures." This issue needs clarification.
Also, the physiological mechanisms linking epileptic seizures and HRV changes should be at least briefly mentioned in the Discussion section.
At the same time, I believe it would be interesting and rational to study not only HRV and HR, but also ECG phenomena such as arrhythmias and, ultimately, small changes in the shape of the electrocardiographic curve itself. Perhaps this aspect should be included in the "Further Research" section.
In conclusion, I would like to emphasize once again that in general the article is of significant scientific interest and deserves publication.
Reviewer 3 Report
Comments and Suggestions for Authors
Overall, the manuscript is well-written, and the idea is interesting and original. The plagiarism rate was analyzed and found to be 7%, which is acceptable. Some suggestions were simple and others a bit more complex. However, after addressing these suggestions, the manuscript will be suitable for publication.
TimeVQVAE-AD: The full method name is initially introduced consistently as TimeVQVAE-AD in the Abstract (line 12) and again in the results (line 14). However, the tables present it in a mix, sometimes as Time-VQVAE-AD (Table 1, line 416). Please unify the terminology to one term throughout the manuscript.
Matrix Profile / Matrix Profiling: The name of this method is inconsistent, as can be seen in line 12 and in Table 1, line 416. Please unify the terminology to one term throughout the manuscript.
Figure A1 Caption: The current caption is too brief (line 886). Revise the caption to be fully informative for a clear comparison.
To our knowledge, no study has yet evaluated ECG-based seizure detection methods on this dataset. (line 60). This statement has no scientific basis. To indicate that there are no studies on a specific topic, it is necessary to describe the search process and mention the databases that were used. In order to clarify what is meant by “to our knowledge.”
Review reference format [10, p.1]. on line 58
Lines 74-94 contain results, discussions, and conclusions. This information should be included in the Results, Discussion, and Conclusions sections, respectively.
Line 135. This sentence is not clear. It is not clear whether this scenario (50 bpm change) occurs 55.55% of the time when the patient presents the first seizure, or if when this scenario occurs, 55.55% of the time the first seizure occurs.
Reviewer 4 Report
Comments and Suggestions for Authors
The article “ECG-Based Detection of Epileptic Seizures in Real-World Wearable Settings: Insights from the SeizeIT2 Dataset” by Reintjes et al. demonstrates the feasibility of ECG-only seizure detection by using the SeizeIT2 dataset. It introduced three methods and compared their performance with different optimized metrics. Though the result is promising and the article is well-written, the article lacks figure demonstration, especially description of the three approaches simply by text is hard to follow. Please add corresponding figures to illustrate the process better. A major revision is suggested at this moment, other comments see below
- Clinically, invasive vagus nerve stimulation (VNS) is an FDA-approved approach for treating drug-resistant sepsis, and there are FDA-approved closed-loop VNS devices that monitor patients’ heartbeats and automatically deliver vagal stimulation when they detect an abnormal change1. This is worth mentioning and discussing in the introduction.
- Line 147, does the author mean to downsample the ECG signal to 8Hz? Will that be too aggressive and miss important cardiac information?
- The article uses sub-097 to sub-125 as the test set. While this patient-level holdout prevents data leakage, the single fixed split limits generalizability assessment and provides no statistical comparison between methods. Consider using k-fold cross-validation and test on different test sets, and perform statistical comparison between models. Additionally, demonstrating results in a figure format will be more straightforward than a table.
- What’s the processing time for each approach? This is important information for real-time epilepsy detection
- In the discussion, the authors discussed other works using an EEG-based approach from the same dataset. It would be more straightforward to add figures such as bar plots to directly show the comparison.
- Please make sure have units for all result (e.g. Line 264-266)
1 Afra, P., Adamolekun, B., Aydemir, S. & Watson, G. D. R. Evolution of the vagus nerve stimulation (VNS) therapy system technology for drug-resistant epilepsy. Frontiers in Medical Technology 3, 696543 (2021).
Round 2
Reviewer 1 Report
Comments and Suggestions for Authors
Authors have made substantial improvements that significantly strengthen the scientific rigor and clarity of your work. The addition of pipeline figures, statistical analyses, and seizure-type evaluations greatly enhances the manuscript's quality. The detailed responses in the cover letter is really appreciated.
However, some minor revisions were identified and should be addressed as follows:
- The p-value column in Tables 2 and 3 requires clarification. Currently, it is not immediately clear which comparison each p-value represents. Furthermore some p-values reported as "0.0000" which are not correct; they should follow standard statistical conventions as "<0.0001".
- Some seizure types have very small sample sizes (e.g., ua_nm: n=4), which limits the reliability of type-specific sensitivity estimates.
- While the clinical translation discussion was expanded, the practical integration pathways could be more concrete.
- About the HRV-based method comparison, the response to Comment 3 appropriately explains why direct comparison with Jeppesen's method is not feasible. However, this important point should be more explicitly stated in the discussion to guide future research.
Once these minor points are addressed, the manuscript will be suitable for publication. The scientific contribution is significant by providing the first comprehensive benchmark for ECG-based seizure detection on SeizeIT2, with statistical analysis and clinically relevant evaluation frameworks. The work will be valuable to the seizure detection research community.
Reviewer 4 Report
Comments and Suggestions for Authors
The paper has been significantly improved after the revision. It maybe accepted in its current form.
Author Response
Thank you very much for reviewing our revised manuscript and for the constructive comments provided in the first round. We are pleased to hear that the paper has significantly improved and appreciate the reviewer’s positive assessment in this second round. We are grateful for the time and effort invested in evaluating our work and are glad that the manuscript is now considered suitable for acceptance. Should any further adjustments be required, we will be happy to address them.